# Early Mediterranean-Based Nutritional Intervention Reduces the Rate of Gestational Diabetes in Overweight and Obese Pregnant Women: A Post-Hoc Analysis of the San Carlos Gestational Prevention Study

**DOI:** 10.3390/nu16142206

**Published:** 2024-07-10

**Authors:** Rocío Martín-O’Connor, Ana Ramos-Levi, Veronica Melero, María Arnoriaga-Rodriguez, Ana Barabash, Johanna Valerio, Laura del Valle, Paz de Miguel, Angel Diaz, Cristina Familiar, Inmaculada Moraga, Alejandra Duran, Martín Cuesta, María José Torrejón, Mercedes Martínez-Novillo, Clara Marcuello, Mario Pazos, Miguel A. Rubio, Pilar Matía Matin, Alfonso L. Calle-Pascual

**Affiliations:** 1Endocrinology and Nutrition Department, Hospital Clínico Universitario San Carlos, Instituto de Investigación Sanitaria del Hospital Clínico San Carlos (IdISSC), 28040 Madrid, Spain; rmoconnor@salud.madrid.org (R.M.-O.); ana_ramoslevi@hotmail.com (A.R.-L.); veronica.meleroalvarez10@gmail.com (V.M.); maria.arnoriaga.rodriguez@gmail.com (M.A.-R.); ana.barabash@gmail.com (A.B.); valeriojohanna@gmail.com (J.V.); lauradel_valle@hotmail.com (L.d.V.); pazdemiguelnovoa@gmail.com (P.d.M.); joseangel.diaz@salud.madrid.org (A.D.); cristinafamiliarcasado@gmail.com (C.F.); inmamgg@hotmail.com (I.M.); aduranrh@gmail.com (A.D.); cuestamartintutor@gmail.com (M.C.); clara994@hotmail.com (C.M.); mario.pazos@salud.madrid.org (M.P.); marubioh@gmail.com (M.A.R.); 2Facultad de Medicina, Medicina II Department, Universidad Complutense de Madrid, 28040 Madrid, Spain; 3Centro de Investigación Biomédica en Red de Diabetes y Enfermedades Metabólicas Asociadas (CIBERDEM), 28029 Madrid, Spain; 4Clinical Laboratory Department Hospital Clínico Universitario San Carlos, Instituto de Investigación Sanitaria del Hospital Clínico San Carlos (IdISSC), 28040 Madrid, Spain; mariajosefa.torrejon@salud.madrid.org (M.J.T.); mercedes.martineznovillo@salud.madrid.org (M.M.-N.)

**Keywords:** obesity, overweight, metabolic syndrome, fasting blood glucose, Mediterranean diet, gestational diabetes

## Abstract

Obesity is a risk factor for the development of gestational diabetes mellitus (GDM). However, the most optimal type of nutritional intervention to prevent GDM in high-risk women is not clearly defined. This study investigates if nutritional treatment based on the Mediterranean diet (MedDiet) before the 12th gestational week (GW) in women at high risk due to a body mass index (BMI) ≥ 25 kg/m^2^ reduces the rate of GDM and metabolic syndrome (MetS) at 3 years postpartum. We performed a post-hoc analysis of the San Carlos Gestational Prevention Study. A total of 735 women with BMI ≥ 25 kg/m^2^ were evaluated between 2015 and 2018, with 246 in the standard diet control group (CG) and 489 in the MedDiet intervention group (IG). The rate of GDM was significantly lower in IG compared to CG (25.1% vs. 31.7%), relative risk (95% confidence interval), and 0.89 (0.78–0.99); *p* = 0.037. Postnatal follow-up was completed by 141 women in CG (57%) and 312 women in IG (64%). At 3 years postpartum, we observed a reduction in the rates of impaired fasting glucose (IFG) (0.51 (0.28–0.92); *p* = 0.019), obesity (0.51 (0.28–0.92); *p* = 0.041), waist circumference (WC) ≥ 89.5 cm (0.54 (0.31–0.94); *p* = 0.022), and MetS (0.56 (0.33–0.94); *p* = 0.003). MedDiet reduces the rate of GDM and postpartum MetS in women with BMI) ≥ 25 kg/m^2^, suggesting that its implementation should be routinely recommended from the first GWs.

## 1. Introduction

The prevalence of people being overweight or obese is a growing public health problem worldwide [1,2]. This pathophysiological condition is associated with metabolic disturbance and increased insulin resistance [3], as well as other pregnancy complications such as preterm birth, hypertensive disorders, and cardiovascular diseases [4,5,6]. Moreover, during mid-pregnancy, placental hormones change maternal physiology to achieve a state of insulin resistance to support fetal growth. Thus, starting pregnancy with pre-existing obesity-related insulin resistance poses an increased risk of developing Gestational Diabetes Mellitus (GDM) [7]. It is well documented that both GDM and obesity are independently associated with adverse maternal and infant outcomes such as pre-eclampsia, emergency caesarean delivery, fetal macrosomia and neonatal hypoglycemia [8,9,10,11]. Previous studies in this healthcare area have observed a significant increase in the risk of GDM, with its associated maternal and fetal adverse events [12,13]. Consequently, the development of preventive strategies is a priority.

Numerous randomized controlled trials (RCTs) have been conducted to attempt a reduction in the incidence of GDM. These trials have explored the effect of dietary modifications, physical activity, combined interventions, and medication. However, the results remain controversial [14], and very few have demonstrated efficacy in high-risk populations [15,16,17]. Heterogeneities of the population including, the types of intervention evaluated and the time of their initiation are some of the main causes that explain the inconsistencies observed [18]. The treatment of GDM has focused on proper glycemic control and adequate weight gain during pregnancy using diet and exercise [7,18,19], and, if lifestyle approaches alone are not sufficient, insulin or other antihyperglycemic drug therapies may be prescribed [20]. Nutritional intervention may include a wide range of possibilities. Although no specific diet has been described for the management of GDM, previous studies have shown that large amounts of carbohydrates (CHO), especially rapidly absorbed CHO, have a negative impact on glycemic control [21,22,23,24]. Therefore, individualized moderation of CHO intake is reasonable, and the focus should turn to the type of CHO, prioritizing those high in fiber.

The mediterranean diet (MedDiet) is high in complex CHO and healthy fats. It has demonstrated many health benefits for people with obesity, including weight loss and a reduction in associated comorbidities [25,26,27]. In this regard, the San Carlos study has demonstrated a decrease in GDM in its cohort of pregnant women using a prompt (12th gestational week -GW-) nutritional intervention, based on a MedDiet, with a free supply of extra virgin olive oil (EVOO) and nuts [13]. However, prevention strategies in the specific setting of overweight and obesity has not been fully explored.

In this context, the main aim of this study is to investigate if early nutritional treatment in overweight or obese women reduces the incidence of GDM and its associated complications, including maternal-fetal adverse effects during pregnancy and at 3 years postpartum.

## 2. Materials and Methods

### 2.1. Study Design

The Hospital Clínico San Carlos is a recognized healthcare, education and research center in Madrid, Spain. Pregnant women have their first clinical visit between the 8th and 12th GW, coinciding with the first ultrasound scan. At this visit, the first trimester’s blood tests are reviewed, and general recommendations are given for a good pregnancy. The oral glucose tolerance test (OGTT) to detect GDM is performed between 24–28 weeks of gestation, using the IADPSG criteria. With the aim of preventing GDM, the San Carlos study, a single-center, prospective, RCT, was launched in 2015. It included all pregnant women attending their first gestational visit between the 8th and 12th GW with fasting blood glucose (FBG) < 92 mg/dL. This cohort encompasses three main research studies that we have used in our analysis, including only women with a body mass index (BMI) ≥ 25 kg/m^2^ from each of the three cohorts/studies. The Appendix A shows a flow chart of the patients distributions since the inclusion visit.

The first study (Study 1) was a RCT (ISRCTN84389045, ethic code CI 13/296-E), in which women were randomly assigned by age, parity, ethnicity and BMI by an external investigator, before the 12th GW to either the control group (CG) or the intervention group (IG). In this way, nutritional intervention could be ensured for at least 3 months. The IG recommended a MedDiet guideline: insisting on increased consumption of EVOO (>40 mL/day) and pistachios (1 handful per day), with both foods provided free of charge. In contrast, women in the CG group were advised to limit their daily EVOO intake to less than 40 mL and to avoid eating nuts more than three times a week.

The second study (Study 2) (ISRCTN13389832, ethic code CI 16/442-E) assessed the effects of giving MedDiet-based recommendations in early pregnancy in a single group based on usual clinical practice, Real World (RW). These women were encouraged to follow the same nutritional guidelines as the RCT IG of Study 1, but without providing them with free EVOO and nuts.

The last study (Study 3) was an RCT (ISRCTN16896947, ethic code CI 16/316) in which women with a body mass index (BMI) ≥ 25 kg/m2 were randomized into CG and IG by age, parity, ethnicity and BMI by an external investigator. In this case, the intervention consisted of increasing the consumption of nuts and EVOO, but only pistachios were administered free of charge, not EVOO.

A post hoc analysis of this sample was performed for women in either overweight or obesity (BMI ≥ 25 kg/m^2^) categories. IGs from both RCTs and the group participating in the practice-based study underwent the analysis as IGs. This was because they were all recommended to adhere to the principles of the MedDiet based on the use of EVOO as the only fat, in an amount greater than 4 tablespoons per day, and the consumption of a handful of nuts per day, in particular pistachios, uniformly in each study. In study 1, EVOO and pistachios were provided free of charge (10 L EVOO and 2 kg of roasted pistachios at 12–14 and 24–28 GW); whereas in study 3 only 2 kg of pistachios were provided free of charge at 12–14 and 24–28 GW. In study 2, only their consumption was recommended, but no food was provided free of charge. In contrast, women in the CG, both in study 1 and study 3, received identical recommendations to reduce oil consumption to less than 4 tablespoons per day, without the need of EVOO exclusively, and to avoid the consumption of nuts. The rest of recommendations were similar in both groups, such as: consuming 5 servings per day of fruit and vegetables, 2 or 3 servings of dairy products and prioritizing the consumption of whole grains, legumes, fish and lean meats. In addition, reducing the consumption of ultra-processed snacks, processed meats, commercial sweets and soft drinks was emphasized. It was also recommended to maintain an active lifestyle and use water as the main beverage.

IGs were compared with the two CGs in the RCTs. Women diagnosed with GDM were closely monitored by the Department of Endocrinology and received a consistent protocolized treatment, regardless of whether they were assigned to the control or intervention group.

During the study, pregnant women were followed-up uniformly to reinforce the nutritional intervention. Three visits were made during pregnancy, coinciding with the third, fifth and ninth months of gestation. At 3 months postpartum, another motivational interview was conducted to encourage all patients, regardless of the group to which they belonged, to follow the nutritional recommendations to liberalize the consumption of oil, preferably EVOO, and nuts, preferably pistachio, on a daily basis. After 3 years of the study, the patients were invited again for a voluntary follow-up visit to evaluate the results. However, several women refused to participate for different reasons, such as being pregnant again or changing hospital address, among others.

### 2.2. Patients

A total of 735 pregnant women: 246/CG (144/ Study 1 + 102/ Study 3) and 489/IG (120 Study 1 + 162 Study 2 + 207 Study 3) were included in the present study. All agreed to participate in the study and signed the informed consent. Women’s demographic and clinical characteristics at the baseline can be seen in Table 1.

### 2.3. Studied Variables

#### 2.3.1. Clinical and Demographic Characteristics

At the first visit, a brief medical history was taken, including age, ethnicity (Caucasian, Latin American or others), family history of Type 2 diabetes mellitus (T2DM), having more than two components of the MetS (BMI ≥ 30 kg/m^2^, WC ≥ 89.5 cm, systolic blood pressure (SBP)≥ 130 mm Hg, diastolic blood pressure (DBP) ≥ 85 mmHg, triglycerides (TG) ≥ 150 mg, HDL-Chol < 50 mg/dL, abnormal glucose regulation (AGR) and homeostasis assessment model for insulin resistance (HOMA-IR) ≥ 3.5), previous miscarriages, educational level (elementary education, secondary school, and university degree), employment status, number of previous pregnancies and smoking habits (never or currently). Anthropometric data such as weight (kg) and BMI (kg/m^2^) were also collected, and blood pressure was measured using a digital sphygmomanometer (Omron 705IT, Omron Global, Kyoto, Japan). In patients who were followed up at 3 years postpartum, it was possible to analyze body composition by means of electrical bioimpedance (SECA mBCA 514), obtaining weight (kg), fat mass (FF) (kg) and BMI (kg/m^2^). WC was measured with a non-stretch tape measure using ISAK criteria.

#### 2.3.2. Laboratory Parameters

Blood and urine samples were obtained after an overnight fast of at least 8 h. Fasting blood glucose (mmol/L) was measured in serum by the glucose-hexokinase method in an AU5800-Beckman Diagnostic. To measure HbA1c as a percentage, ion exchange high-performance liquid chromatography (HPLC) was perfomed with a Tosoh G8 analyzer (Tosoh Co., Tokyo, Japan). The method is standardized against the International Federation of Clinical Chemistry and has an imprecision of 1.23% for values of 32.23 mmol/mol (5.1% NGSP) and 1.36% for values of 85.24 mmol/mol (10% NGSP). To measure fasting serum insulin (FSI), a chemiluminescence immunoassay on an IMMULITE 2000 Xpi (Siemens, Healthcare Diagnostics, Munich, Germany) was used, with an imprecision of 21 IUU/mL concentrations of 5.91%. Homeostasis for insulin resistance (HOMA) was calculated as glucose (mmol/L) × insulin (μIU/mL)/22.7. Total cholesterol was quantitatively assessed using the colorimetric enzymatic test method (CHOD-PAP). In a similar way, the colorimetric enzymatic method using glycerol phosphate oxidase p-amino phenazone (GPO-PAP) was used to determine serum TG. Fasting serum insulin (FSI) levels were determined using a chemiluminescence immunoassay performed on an IMMULITE 2000 Xpi analyzer from Siemens Healthcare Diagnostics in Munich, Germany. The inter-assay accuracy was 6.3% for concentrations of 11 uIU/mL and 5.91% for insulin concentrations of 21 uIU/mL. Thyroid-stimulating hormone (TSH) levels were assessed using a third-generation sandwich-chemiluminescence immunoassay with magnetic particles. This method employed human TSH mouse monoclonal antibodies and the DXI-800^®^ analyzer from Beckman–Coulter (Brea, CA, USA). The manufacturer’s specified reference range for TSH in non-pregnant adults is 0.38–5.33 μIU/mL, with a sensitivity of 0.01 μIU/mL. The intraassay coefficient of variation (CV) is less than 10%, and the TSH measurement range spans from 0.01 to 50.0 μIU/mL. Intra-assay CVs are 4.9% for a concentration of 0.69 μIU/mL, 5,8% for 5.47 μIU/mL, and 6.2% for 29 μIU/mL. In addition, to ensure the quality of the procedures, an External Quality Assurance Programme of the SEQC (Spanish Society of Clinical Chemistry) evaluates the methods monthly and performs a review of the methods.

#### 2.3.3. Lifestyle and Data Collection

The Mediterranean diet adherence screener (MEDAS), a validated semiquantitative questionnaire derived from the PREDIMED study, was used to evaluate the nutritional intervention and adherence to the MedDiet [28]. A score above 5 indicated adequate adherence, although the aim was to achieve a minimum score of 7. The second questionnaire applied was the Nutrition and Diabetes Complications Trial (DNCT), which more precisely reflects the consumption of specific food groups per week, as well as the physical activity carried out on a regular basis. This questionnaire evaluates 15 items: 3 of the m focused on physical activity and the remaining 12 on diet. Factors that prevent T2DM were given an A (+1 point); neutral factors were given a B, which do not prevent T2DM but do not increase the risk (+0 points); and a C was given when the factor increased the risk of developing T2DM (−1 point). Further details of the assessment are given in the pilot study [29]. They were applied at each of the visits made by the patients.

### 2.4. Statistical Analysis

Categorical variables are represented as percentages, while continuous variables are expressed as median (standard deviation). Differences between groups were compared using X2 tests, Student’s *t*-test, or the Mann-Whitney U-test, depending on the type of variable and its normal or non-normal distribution. Logistic regression was used to assess the impact of nutritional treatment on adverse maternal, neonatal and delivery outcomes that showed significant differences in univariate analysis. The control group served as the reference group. Crude relative risk (RR) and 95% confidence intervals were calculated. All *p*-values were two-tailed and were considered statistically significant if less than 0.05. These analyses were performed using SPSS software, version 21, based in Chicago, IL, USA.

## 3. Results

Table 2 shows maternal and fetal outcomes during pregnancy in the CG and IG. The rate of GDM was lower in the IG (25.1%) in comparison to the CG (31.7%) and RR (95% CI) 0.89 (0.78–0.99) *p* = 0.037, which was associated with lower fasting basal glucose. HbA1c was also lower in both the second and third trimesters of pregnancy (5.1 vs. 4.9; *p* = 0.001 and 5.4 vs. 5.2; *p* = 0.001, respectively), although neither value conferred a diagnosis of pre-diabetes (HbA1c ≥ 5.7%). No significant differences were observed in weight gain during pregnancy or adequate weight gain from baseline BMI. In terms of treatment, no significant changes were observed in the number of patients who had nutritional intervention or in total insulin doses between the two groups. However, there were differences in basal insulin requirements, with lower doses needed in IG versus CG (11.5% vs. 0.1%; *p* = 0.030), probably due to lower fasting basal glucose levels. IG women had lower rates of other associated comorbidities such as diastolic blood pressure (DBP) between the 36 and 38th GW, and urinary tract infection (UTI) and albuminuria. There were no differences in blood pressure between the 24 and 28th GW, in the diagnosis of hypertension or preeclampsia, or in bacteriuria. Regarding neonatal outcomes, there was an increase in small-for-gestational-age (SGA) neonates in the IG relative to the CG (6.1 vs. 2.8%; *p* = 0.036). However, there were no differences in other anthropometric variables, such as weight and length, nor in large-for-gestational-age (LGA) infants. There were also no differences in rates of prematurity or dystocia, umbilical cord pH, APGAR, or other adverse effects in the newborn (hypoglycemia, respiratory problems, hyperbilirubinemia).

Table 3 shows biochemical, anthropometric, and clinical data of women at postnatal, 3-month and 3-year follow-up. This follow-up was completed by a total of 141 women of the CG women (57%) and 312 of the IG women (64%). We found a higher weight gain at 3 years postpartum in the CG compared to the IG (5.1 ± 10.4 vs. 3.6 ± 6.5; *p* = 0.045). There were no significant differences in other anthropometric measures. In analytical parameters, no differences were seen except for higher LDL-Chol in CG against IG at 3 months postpartum (128 ± 29 vs. 118 ± 31; *p* = 0.007). In terms of lifestyle, differences in MedDiet adherence were observed, with higher scores in IG than in CG at both 3 months and 3 years postpartum (6.4 ± 1.7 vs. 5.9 ± 1.9; *p* = 0.042 and 6.9 ± 2.0 vs. 6.5 ± 1.9; *p* = 0.013). However, no differences were found in the level of physical activity.

A detailed comparison of the postpartum rate of MetS components between overweight and obese women separated by IG vs. CG and women with GDM vs. normal glucose tolerance (NGT) is shown in Table 4. We compared the relative risk (RR) of developing MetS and its components according to the nutritional intervention at 3 months (panel A) and 3 years (panel B) after delivery. No significant differences were found at 3 months postpartum between IG and CG in any of the MetS components. However, as expected, at 3 months postpartum there was a reduction in RR (95% CI) in the GDM group versus the NGT group in the rates of IFG, prediabetes, and all MetS components (WC ≥89.5 cm, SBP of ≥130 mmHg, DBP ≥ 85 mmHg, and AGR), except for HOMA-IR in which there were no differences between groups.

In contrast, at 3 years postpartum (panel B) there was a reduction in the RR (95% CI) of the IFG rate (0.51 (0.28–0.92); *p* = 0.019). Also, there were significant differences in the risk of developing MetS between both groups. This would imply that the nutritional intervention could be a protective factor against long-term MetS, as there was a higher rate of women with obesity in the CG (0.51 (0.28–0.92); *p* = 0.041), as well as a higher rate of women with a WC ≥89.5 cm (0.54 (0.31–0.94); *p* = 0.022). There were no significant differences for the other components of the MetS or in the RR of developing prediabetes.

Finally, comparing the GDM group with the group of women with NGT at 3 years postpartum, an increase in the RR of both IFG (1.84 (1.34–2.53); *p* = 0.000) and prediabetes (1.73 (1.15–2.60); *p* = 0.001) was observed. Regarding the components of MetS, a higher risk of developing MetS for having more than 2 components was found in the GDM group vs. women with NGT (1.62 (1.01–2.65); *p* = 0.008). This was due to the increased rate of women with WC ≥89.5 cm and AGR. On the contrary, there was no significant difference in the rate of women with BMI > 30 kg/m^2^, HOMA-IR ≥ 3.5, elevated blood pressure (SBP ≥130 mmHg, nor DBP ≥ 85 mmHg), or a dysregulated lipid profile (TG ≥150 mg/dL and HDL < 50 mg/dL) between both groups.

## 4. Discussion

This study shows that a nutritional intervention based on MedDiet significantly reduces the rate of GDM and the RR of developing MetS at 3 years postpartum, when implemented at the beginning of pregnancy (before 12 weeks of gestation) in women with BMI ≥ 25 kg/m^2^. This is because FBG and HBA1c levels were significantly lower in the IG group.

A recent systematic review suggested that nutritional intervention during pregnancy and postpartum may improve glucose regulation in patients with GDM [30]. Similarly, this retrospective study points to the importance of personalized dietary intervention to reduce adverse events in pregnancy for both mother and child [31]. However, nutritional intervention covers a wide range of possibilities. This retrospective study advocates a personalized diet, concluding that this helps to stabilize blood glucose levels [32]. Furthermore, in high-risk women there is not much literature, but it is known that this condition poses a higher risk of maternal-fetal complications [33]. This is why many studies insist on focusing on the type of nutritional intervention recommended, and aim to develop specific guidelines for a standardized approach [7,34,35]. Our study has provided evidence for using a nutritional intervention with MedDiet based on vegetables, fruits, legumes, whole grains, lean meats instead of processed meats, oily fish, EVOO and nuts, as well as for reducing consumption of processed and sugary foods as early as possible in pregnant women to reduce the likelihood of developing GDM and the long-term negative postnatal metabolic impact. In addition, the dietary recommendations given are specific, measurable, and assessed through validated semi-quantitative questionnaires. This allows us to not only reproduce the study, but also to establish specific food consumption frequencies for dietary recommendations.

We found no significant differences in the rate of maternal-fetal adverse events. This may be because the total sample included women who are already at high prior risk of developing adverse events in pregnancy due to their own excess weight. In fact, all patients studied presented a BMI ≥ 25 kg/m^2^ both at baseline and after intervention. A recent RCT in pregnant women at risk of hyperglycemia [35] showed a significant, but modestly smaller, decrease in adverse neonatal events. This study performed the intervention later, before 20 weeks’ gestation, and has the limitation of non-standardized treatment of GDM. Perhaps these differences are explained by the fact that this study included overweight patients with additional risk factors for developing GDM, such as a previous diagnosis of GDM, high gestational age, first-degree relatives with diabetes, polycystic ovarian syndrome, and non-European ancestry. In addition, the sample included a limited number of Hispanic women.

Previous studies have focused on the beneficial effects of the prevention of GDM on the control of total weight gain during pregnancy [7,34,36,37]. However, in this study, we observed a reduction in the rate of GDM, even in the setting of no significant differences in total weight gain between women in the GC and the IG. This could be because of the influence of the complex interrelationship between weight, fat distribution and diet. In fact, we did observe a higher rate of women with WC ≥ 89.5 cm at 3 years postpartum in the CG, drawing the attention to the importance of body composition and distribution in the follow-up of weight gain during pregnancy. Thus, our study reinforces what other large studies such as PREDIMED have shown: MedDiet prevents the development of MetS in the long-term follow-up due to a lower central distribution of adipose tissue. It would be interesting to validate methods for the assessment of body composition in pregnant women, in order to develop more personalized nutritional recommendations in the specific setting of pregnancy [38].

A topic of great interest concerns the timing of initiation and duration of the nutritional intervention set for pregnant women for the prevention of GDM. Many studies argue in favor of its implementation as soon as possible, especially in high-risk patients [37,39]. In this regard, previous reports suggest the need for initiating the intervention in the 20th GW to achieve a relevant reduction in GDM (29), but not all reports have observed a clear benefit when implementing it at this time [15]. There is no doubt that appropriately designed and targeted interventions can be effective tools for the management of pregnancies that are complicated by overweight and obesity. However, the demonstration of its efficacy is somehow blurred because of a lack of resources and the failure of previous studies to establish the appropriate time at which this personalized nutritional care should be provided to effectively prove a benefit. Thus the controversy continues. However, the need for a lifestyle intervention for the prevention of postnatal GDM and T2DM is beyond doubt [40,41,42].

Benefits of Mediterranean diet supplemented with EVOO and pistachios may be due to several mechanisms. For instance, it has been observed that EVOO reduces postprandial glucose levels and improves the inflammatory profile, and its use facilitates a greater intake of vegetables, traditionally consumed with EVOO in Spanish cuisine. It seems possible that it increases satiety due to the fact that fats delay gastric emptying and also thermogenesis. On the other hand, nuts, and in particular pistachios, are rich in unsaturated fatty acids, fiber, magnesium and other phytochemical components with possible beneficial effects on insulin sensitivity, fasting glucose levels and inflammation. Their antioxidant capacity is superior to other nuts, given their high levels of lutein, β-carotene and γ-tocopherol. Pistachio consumption improves inflammatory cytokine profiles related to the development of GDM [43,44,45].

Our study entails some limitations. Firstly, this is a post-hoc study, meaning that the initial main objective was not to evaluate women who are overweight or obese. This limits the sample size; however, we have managed to group enough patients to obtain significant data. On the other hand, postnatal follow-up was performed in approximately 60% of women, mainly due to the difficulty of attendance because of the recent delivery, having other small children, and unavailability to attend medical follow-ups. Other causes were a change in community, follow-up in a private hospital or not wishing to continue participating in the study. However, this proportion is still considered acceptable and is descriptive of our population, representing more than half of the total sample. This highlights the challenge of adhering to the recommended postnatal follow-up by all scientific guidelines. Another limitation is that diet is a subjective, complex, and changing parameter, also questionnaire responses could be biased. However, semi-quantitative questionnaires are a standardized, quick, and non-invasive tool to adequately represent food intake, and are always administered by specialized professionals to minimize bias. Therefore, the use of validated questionnaires allows us to overcome this limitation, and provides a genuine character to our study, by objectively quantifying and measuring the nutritional intervention carried out.

## 5. Conclusions

In conclusion, our research demonstrates that an early nutritional intervention using the MedDiet, along with supplementation of EVOO and nuts, in women with a BMI ≥ 25 kg/m^2^ and initiated during early pregnancy, decreases the rate of GDM and positively influences the risk of developing MetS over a three-year period after delivery. Therefore, indicating MedDiet-based dietary patterns in women with excess weight from early pregnancy can be considered a preventive strategy for the development of postnatal GDM and MetS. Future studies with a defined and measurable nutritional intervention in high-risk women are needed to further confirm our findings.

## Figures and Tables

**Table 1 nutrients-16-02206-t001:** Characteristics of the maternal population with a pregnancy BMI ≥ 25 kg/m^2^ at the baseline visit (8–10th GW) assessed in the clinical trial population by groups.

	Control (N = 246)	Intervention (N = 489)	*p*
Age (years)	31.8 ± 5.6	32.9 ± 5.1	0.067
Race/Ethnicity			
Caucasian	121 (49.6)	267 (54.7)	
Latin American	117 (48.0)	206 (42.2)	
Others	8 (2.5)	16 (3.1)	0.103
Family history of T2D	62 (25.3)	155 (31.3)	
MetS (>2 components)	56 (22.9)	114 (23.4)	0.330
Previous history of GDM	7 (2.8)	24 (4.7)	
Miscarriages	93 (37.9)	195 (39.9)	0.137
Educational status:			
Elementary education	32 (13.1)	37 (7.6)	
Secondary School	93 (38.0)	162 (33.2)	
University Degree	117 (47.8)	278 (56.9)	
UNK	4 (1.2)	11 (2.2)	0.066
Employment	187 (76.3)	376 (77.0)	0.355
Number of pregnancies:			
Primiparous	90 (36.7)	171 (35.1)	
Second pregnancy	66 (26.9)	154 (31.6)	
>2 pregnancies	90 (36.8)	164 (33.4)	0.608
Smoker:			
Never	152 (62.0)	269 (55.1)	
Current	14 (5.7)	42 (8.6)	0.382
Body Weight (kg):			
Pre pregnancy	72.7 ± 12.1	71.8 ± 11.2	0.336
At baseline	75.5 ± 11.6	74.8 ± 10.4	0.365
BMI (kg/m^2^):			
Pre pregnancy	27.8 ± 3.9	27.4 ± 3.6	0.223
At baseline	28.9 ± 3.7	28.6 ± 3.3	0.209
Blood Pressure (mmHg):			
Systolic	113 ± 9	113 ± 11	0.438
Diastolic	69 ± 9	70 ± 9	0.484
FBG (mmol/L)	4.5 ± 0.4	4.5 ± 0.3	0.380
Insulin	22 ± 26	23 ± 26	0.567
HOMA-IR	1.4 ± 1.8	1.5 ± 1.7	0.587
Cholesterol (mg/dl)	175 ± 33	179 ± 33	0.330
Triglycerides	94 ± 46	93 ± 48	0.806
HbA1c %	5.1 ± 0.2	5.1 ± 0.3	
mmol/mol	32 ± 0.8	32 ± 0.9	0.928
TSH mcUI/mL	2.1 ± 1.3	2.1 ± 1.6	0.482
MEDAS Score	4.7 ± 1.7	4.9 ± 1.7	0.221
Nutrition Score	0.2 ± 3.2	0.2 ± 3.1	0.859
Physical Activity	−1.8 ± 0.9	−1.8 ± 0.9	0.872

Data are Mean + SD or number (%). Type 2 Diabetes (T2D); Metabolic Syndrome (MetS); Gestational Diabetes Mellitus (GDM); Unknown (UNK); Body Mass Index (BMI); Fasting Blood Glucose (FBG); Homeostasis assessment model for insulin resistance (HOMA-IR); Mediterranean Diet Adherence Screener (MEDAS). Physical Activity (PA) Score, (Walking daily (>5 days/week) Score 0: At least 30 min. Score +1, if >60 min. Score −1, if <30 min. Climbing stairs (floors/day, >5 days a week): Score 0, between 4 and 16; Score +1, >16; Score −1: <4).

**Table 2 nutrients-16-02206-t002:** Maternal, pregnancy, and neonatal outcomes.

*Maternal Outcomes*	Control (N = 246)	Intervention (N = 489)	*p*
GDM n (%)	78 (31.7)	123 (25.1)	**0.037**
75 g-OGTT 24–28 GW			
FBG (mmol/L)	4.9 ± 0.4	4.7 ± 0.4	**0.014**
1 h Blood Glucose mmol/L	7.1 ± 1.7	7.0 ± 1.7	0.839
2 h Blood Glucose mmol/L	6.1 ± 1.5	6.1 ± 1.3	0.885
HbA1c			
24–28 GW%	5.1 ± 0.3	4.9 ± 0.3	**0.001**
mmol/mol	32 ± 0.9	30 ± 0.9	
36–38 GW%	5.4 ± 0.4	5.2 ± 0.3	**0.001**
mmol/mol	34 ± 0.9	33 ± 0.8	
FBG 36–38 GW (mmol/L)	4.5 ± 0.4	4.4 ± 0.4	**0.046**
FSI (mcUI/mL)			
24–28 GW	13 ± 5.7	10 ± 6	**0.040**
36–38 GW	17 ± 12	14 ± 13	**0.037**
HOMA-IR			
24–28 GW	2.8 ± 1.9	2.4 ± 1.5	**0.037**
36–38 GW	4.0 ± 2.7	3.7 ± 5.3	0.085
Nutritional treatment	41 (52.6)	66 (53.7)	
Insulin requirements (total)	37 (47.4)	57 (46.3)	0.192
Bolus	6 (7.7)	4 (3.2)	
Basal	22 (28.2)	52 (42.2)	**0.030**
Basal/Bolus	9 (11.5)	1 (0.1)	
Weight gain (kg)			
to Baseline (8–10 GW)	2.9 ± 4.3	2.8 ± 4.6	0.817
to 24–28 GW	7.7 ± 5.4	7.3 ± 5.8	0.479
to 36–38 GW	11.1 ± 8.5	11.6 ± 7.9	0.549
Adequate weight gain			
To 24–28 GW (<5 kg)	78 (31.7)	162 (33.1)	0.273
To 36–38 GW (< 9 kg)	66 (26.8)	118 (24.1)	0.798
BP (mm Hg)			
24–28 GW Systolic	110 ± 10	110 ± 10	0.782
Diastolic	66 ± 8	66 ± 8	0.939
36–38 GW Systolic	119 ± 12	117 ± 11	0.069
Diastolic	75 ± 9	72 ± 9	**0.046**
Pregnancy-induced			
Hypertension	13 (5.2)	28 (5.7)	0.902
Preeclampsia	5 (2.0)	13 (2.7)	0.698
Albuminuria	8 (3.3)	4 (0.1)	**0.019**
Bacteriuria	63 (25.6)	105 (21.5)	0.119
Urinary Tract Infection	33 (13.4)	27 (5.5)	**0.001**
Delivery			
Vaginal eutocic	142 (58.1)	277 (56.7)	
Instrumental	45 (18.5)	75 (15.9)	
Cesarean section	59 (23.4)	137 (27.2)	0.579
Emergency	32 (54.3)	59 (43.1)	0.051
*NEONATAL OUTCOMES*			
GW at birth	39.7 ± 1.3	39.4 ± 1.7	**0.032**
≤37 weeks	7 (2.9)	23 (4.7)	0.415
≤34 weeks	1 (0.5)	3 (0.6)	0.108
Birthweight (g)	3302 ± 442	3283 ± 568	0.666
Centile	51 ± 29	54 ± 29	0.303
Height (cm)	49.4 ± 2.0	49.4 ± 2.2	0.970
Centile	43 ± 30	45 ± 28	0.388
LGA > 90 centile	12 (4.9)	37 (7.6)	0.109
SGA < 10 centile	7 (2.8)	30 (6.1)	**0.036**
Ph	7.27 ± 0.06	7.28 ± 0.08	0.161
≤7	12 (0.5)	36 (0.7)	0.783
1 min Apgar	8.8 ± 0.8	8.7 ± 1.2	0.070
<7	4 (1.6)	19 (3.9)	0.669
5 min Apgar	9.9 ± 0.4	9.7 ± 0.8	0.074
<7	3 (1.2)	14 (2.8)	0.496
Neonatal			
Hypoglycemia	6 (2.4)	7 (1.4)	0.384
Respiratory distress	4 (1.6)	7 (1.4)	0.570
Hyperbilirubinemia	7 (2.8)	17 (3.5)	0.422
NICU	12 (4.9)	12 (2.5)	0.108

Gestational Diabetes Mellitus (GDM); 75 g oral glucose tolerance test (75 g-OGTT); Fasting Blood Glucose (FBG); Fasting Serum Insulin (FSI); Homeostasis assessment model for insulin resistance (HOMA-IR); Gestational Week (GW); Blood Pressure (BP); Large for Gestational Age (LGA); Small for Gestational Age (SGA); Neonatal intensive care unit (NICU).

**Table 3 nutrients-16-02206-t003:** Postnatal biochemical, anthropometric, and clinical data of women during postnatal follow-up, at 3 months and 3 years post-delivery by groups.

	CG (N = 141; 57%)		IG (312; 64%)		*p* (CG vs. IG)	
	3-M PD	3-Year PD	3-M PD	3-Year PD	3-M PD	3-Year PD
BW (kg)	77.2 ± 11.1	77.7 ± 16.2	77.1 ± 10.9	75.0 ± 12.9	0.978	0.770
BMI (kg/m^2^)	29.7 ± 3.9	29.3 ± 5.0	29.6 ± 3.8	28.3 ± 3.7	0.840	0.491
BW-Change (kg)	5.8 ± 6.4	5.1 ±10.4	5.2 ± 7.1	3.6 ± 6.5	0.528	**0.045**
WC (cm)	93 ± 8	93 ± 11	93 ± 9	92 ± 10	0.718	0.903
FM (kg)	Na	30.4 ± 12.3	Na	29.4 ± 6.2	---	0.652
sBP (mmHg)	116 ± 15	117 ± 13	115 ± 13	114 ± 12	0.752	0.282
dBP (mmHg)	75 ± 11	75 ± 10	74 ± 10	73 ± 8	0.672	0.336
T-Chol (mg/dL)	203 ± 38	184 ± 29	199 ± 40	179 ±34	0.312	0.284
HDL-Chol	59 ± 12	54 ± 15	60 ± 17	55 ± 11	0.811	0.449
LDL-Chol	128 ± 29	112 ± 25	118 ± 31	107 ± 28	**0.007**	0.116
TG (g/L)	97 ± 50	96 ± 48	95 ± 53	93 ± 45	0.781	0.602
Apo-B (mg/dL)	96 ± 23	90 ± 24	91 ± 25	86 ± 24	0.228	0.496
FSI (μIU/mL)	9.1 ± 8.1	13.2 ± 16.2	8.4 ± 8.3	10.6 ± 7.8	0.485	0.255
HOMA-IR	2.1 ± 1.8	3.9 ± 2.8	2.2 ± 2.8	3.4 ± 4.4	0.634	0.760
FSG (mmol/L)	4.8 ± 0.5	5.1 ± 0.6	4.8 ± 0.4	5.0 ± 0.5	0.623	0.140
2 hOGTT (mmol/L)	Na	5.8 ± 1.7	Na	5.4 ± 0.9	----	0.191
HbA1c-IFCC %	5.3 ± 0.2	5.4 ± 0.3	5.3 ± 0.3	5.4 ± 0.3	0.915	0.728
cPR (mg/dL)	0.42 ±0.49	0.65 ± 1.01	0.62 ± 0.82	0.63 ± 1.04	0.381	0.932
PA Score	−1.6 ± 0.7	−1.8 ± 1.0	−1.7 ± 0.9	−1.6 ± 1.0	0.462	0.305
Nutrition Score	3.5 ± 2.9	1.6 ± 3.1	3.8 ± 3.5	1.8 ± 3.6	0.558	0.710
MEDAS Score	5.9 ± 1.9	6.5 ± 1.9	6.4 ± 1.7	6.9 ± 2.0	**0.042**	**0.013**

Data are Mean (SD). Months (M); Post Delivery (PD); Body weight (BW); Body Mass Index (BMI); Waist Circumference (WC); Fat Mass (FT); No available (Na); systolic blood pressure (sBP); diastolic blood pressure (dBP); total-cholesterol (T-chol.); High-Density Lipoprotein (HDL); Low-Density Lipoprotein (LDL); Triglyceride (TG); lipoprotein B (APO-B); Fasting Serum Insulin (FSI); Homeostasis assessment model for insulin resistance (HOMA-IR); fasting serum glucose (FSG); International Federation of Clinical Chemistry (IFCC); C reactive protein (cPR); Physical Activity (PA) Score, (Walking daily (>5 days/week) Score 0: At least 30 min. Score +1, if >60 min. Score −1, if <30 min. Climbing stairs (floors/day, >5 days a week): Score 0, between 4 and 16; Score +1, >16; Score −1: <4); Mediterranean Diet Adherence Screener (MEDAS).

**Table 4 nutrients-16-02206-t004:** Comparison of the post-delivery rate of metabolic syndrome (MetS) components between obese/overweight women from IG vs. CG and women with GDM vs. a normal glucose tolerance (NGT).

	CG (141) vs. IG (312)		GDM (146) vs. NGT (307)			
	% (N)	RR (95% CI) IG	*p*	% (n)	RR (95% CI) GDM	*p*
*PANEL A. (3 MONTHS)*						
GLYCEMIC STATUS						
IFG	5.0 (7) vs. 4.5 (14)	0.90 (0.36–2.28)	0.496	7.5 (11) vs. 3.3 (10)	1.44 (1.09–2.27)	**0.040**
PREDIABETES (HBA1C ≥ 5.7%)	5.0 (7) vs. 6.7 (21)	1.43 (0.59–3.46)	0.289	13.0 (19) vs. 2.9 (9)	1.47 (1.01–2.13)	**0.008**
METS COMPONENTS						
RAISED (WC ≥ 89.5 CM)	71.3 (100) vs. 68.9 (215)	1.04 (0.87–1.23)	0.332	75.8 (111) vs. 66,5(204)	1.36 (1.19–1.56)	**0.000**
RAISED SBP ≥ 130 MM HG	8.5 (12) vs. 10.5 (33)	1.27 (0.62–2.60)	0.316	19.2 (28) vs. 5.5 (17)	1.22 (1.01–1.56)	**0.041**
RAISED DBP ≥ 85 MM HG	17.0 (24) vs. 14.2 (44)	0.81 (0.41–1.60)	0.326	28.8 (42) vs. 8.5 (26)	1.44 (1.08–1.94)	**0.002**
RAISED TRIG. ≥ 150 MG/DL	11.6 (16) vs. 11.1 (34)	0.96 (0.51–1.79)	0.503	13.8 (20) vs. 9.8 (30)	1.22 (1.00–1.54)	**0.032**
REDUCED HDL-C < 50 mg/dL	21.8 (22) vs. 21.7 (43)	1.00 (0.56–1.78)	0.550	22.3% (21) vs. 21.5% (44)	1.02 (0.84–1.23)	0.217
AGR	7.8 (11) vs. 9.3 (29)	1.25 (0.60–2.61)	0.345	15.1 (22) vs. 5.9 (18)	1.48 (1.11–2.01)	**0.001**
RAISED HOMA-IR ≥ 3.5	12.2 (17) vs. 10.2 (31)	0.81 (0.39–1.71)	0.355	11.6 (17) vs. 10.1 (31)	1.22 (0.93–1.60)	0.069
>2 COMPONENTS OF METS	11.4 (16) vs. 13.2 (41)	1.19 (0.55–2.58)	0.413	17.8 (26) vs. 10.1 (31)	1.37 (1.00–1.96)	**0.035**
*PANEL B. (3 YEARS)*						
GLYCEMIC STATUS						
IFG	17.7 (25) vs. 9.6 (30)	0.51 (0.28–0.92)	**0.019**	21.9 (32) vs. 7.5 (23)	1.84 (1.34–2.53)	**>0.001**
PREDIABETES (HbA1_c_ ≥ 5.7%)	8.5 (12) vs. 8.0 (25)	1.23 (0.57–2.65)	0.370	15.1 (22) vs. 4.9 (15)	1.73 (1.15–2.60)	**0.001**
IGT	9.5 (14) vs. 0	n.a		9.6 (14) vs. 0 (0)	n. a	
METS COMPONENTS						
BMI ≥ 30 (kg/m^2^)	41.1 (58) vs. 24.0 (75)	0.45 (0.19–0.96)	**0.041**	34.2 (50) vs. 27.0 (83)	1.02 (0.76–2.32)	0.547
RAISED (WC ≥ 89.5 cm)	62.5 (88) vs. 44.6 (139)	0.54 (0.31–0.94)	**0.022**	75.0 (102) vs. 40.7 (125)	1.22 (1.07–1.52)	**0.031**
RAISED SBP ≥ 130 mm Hg	13.5 (19) vs. 11.2 (35)	0.90 (0.30–2.73)	0.530	18.5 (27) vs. 8.8 (27)	1.20 (0.75–1.93)	0.298
RAISED DBP ≥ 85 mm Hg	17.0 (24) vs. 5.1 (16)	0.75 (0.51–1.12)	0.227	9.6 (14) vs. 8.5 (26)	1.00 (0.61–1.63)	0.632
RAISED TRIG. ≥ 150 MG/dL	11.3 (16) vs. 10.6 (33)	0.94 (0.45–1.98)	0.507	15.8 (23) vs. 8.5 (26)	1.22 (0.93–1.61)	0.069
REDUCED HDL-C < 50 mg/dL	26.2 (37) vs. 23.4 (73)	0.99 (0.65–1.52)	0.524	43.8 (64) vs. 15.0 (46)	1.12 (0.96–1.31)	0.074
AGR	21.9 (31) vs. 14.1 (44)	0.69 (0.42–1.11)	0.083	28.8 (42) vs. 10.7 (33)	1.72 (1.33–2.23)	**>0.001**
RAISED HOMA-IR ≥ 3.5	12.8 (18) vs. 6.4 (20)	1.00 (0.46–2.18)	0.578	11.6 (17) vs. 6.8 (21)	1.17 (0.84–1.63)	0.227
>2 COMPONETS OF METS	24.1 (34) vs. 8.0 (25)	0.56 (0.33–0.94)	**0.003**	22.6 (33) vs. 8.5 (26)	1.62 (1.01–2.65)	**0.008**

Data are % (number). Control Group (CG); Intervention Group (IG); Gestational Diabetes Mellitus (GDM); Normal Glucose Tolerance (NGT); Impaired Fasting Glucose (IFG); Metabolic syndrome (MetS); Waist Circumference (WC); Systolic Blood pressure (sBP); diastolic blood pressure (dBP); Triglycerides (TRG); High-Density Lipoprotein (HDL); abnormal glucose regulation (AGR); Homeostasis assessment model for insulin resistance (HOMA-IR); Impaired Glucose Tolerance (IGT); Body Mass Index (BMI); Relative Risk (RR); *p* denotes differences between groups.

## Data Availability

The datasets generated during and/or analyzed in the current study are available from the corresponding author upon reasonable request.

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
