# Peer review of "Early Mediterranean-Based Nutritional Intervention Reduces the Rate of Gestational Diabetes in Overweight and Obese Pregnant Women: A Post-Hoc Analysis of the San Carlos Gestational Prevention Study"

_nutrients, 2024, doi:10.3390/nu16142206_

Round 1
Reviewer 1 Report
Comments and Suggestions for Authors
The manuscript by Martín-O´Connor et al. investigates if nutritional treatment based on the Mediterranean diet can lower the risk of developing gestational diabetes in obese women with a BMI >25kg/m2 using multiple comparison tests.
Although this reviewer is not an expert in statistics, could you explain more in the discussion section, taking into account that
the smaller the P value, the more significant the results are, why very little data in table 2, 3 and 4 have a p-value higher than 0,05?
It will be very useful to compare the conclusions of the retrospective studies with those of your prospective study in the discussion section.
The results of a post hoc analysis should be viewed with considerable scepticism, and in my opinion, the degree of novelty regarding the benefits of a Mediterranean diet are well known. It would have been more useful to measure the CHO quantity per day in each patient.
Comments on the Quality of English LanguageEnglish language requires minor corrections.
Author Response
The manuscript by Martín-O´Connor et al. investigates if nutritional treatment based on the Mediterranean diet can lower the risk of developing gestational diabetes in obese women with a BMI >25kg/m2 using multiple comparison tests. Although this reviewer is not an expert in statistics, could you explain more in the discussion section, taking into account that the smaller the P value, the more significant the results are, why very little data in table 2, 3 and 4 have a p-value higher than 0,05?
Thank you very much for your constructive comments with which we agree. We have to make the following considerations:
Rates of GDM are known to be higher in overweight/obese women than in normal weight women. However, a significant reduction in the rate of GDM following lifestyle intervention in this subgroup of patients has not been fully demonstrated. For instance, in the Upbeat study, there were no differences, and in the RADIEL study, only an improvement in glucose regulation parameters was demonstrated, but not in the rate of GDM itself. It is in the ESTEEM study that a significant, albeit small, reduction in the rate of GDM in women with pregestational MetS is obtained, but not all of them were obese.
Here are the references of the studies mention above:
- Poston L, Bell R, Croker H, Flynn AC, Godfrey KM, Goff L, Hayes L, Khazaezadeh N, Nelson SM, Oteng-Ntim E, Pasupathy D, Patel N, Robson SC, Sandall J, Sanders TA, Sattar N, Seed PT, Wardle J, Whitworth MK, Briley AL; UPBEAT Trial Consortium. Effect of a behavioural intervention in obese pregnant women (the UPBEAT study): a multicentre, randomised controlled trial. Lancet Diabetes Endocrinol. 2015 Oct;3(10):767-77. doi: 10.1016/S2213-8587(15)00227-2. Epub 2015 Jul 9. PMID: 26165396
- Koivusalo SB, Rönö K, Klemetti MM, Roine RP, Lindström J, Erkkola M, et al. Gestational Diabetes Mellitus Can Be Prevented by Lifestyle Intervention: The Finnish Gestational Diabetes Prevention Study (RADIEL): A Randomized Controlled Trial. Diabetes Care. enero de 2016;39(1):24-30.
- H Al Wattar B, Dodds J, Placzek A, Beresford L, Spyreli E, Moore A, et al. Mediterranean-style diet in pregnant women with metabolic risk factors (ESTEEM): A pragmatic multicentre randomised trial. PLoS Med. julio de 2019;16(7):e1002857.
To our knowledge, our study is one of the few studies that show benefits of a nutritional intervention (in our case Mediterranean diet) for the prevention of GDM, both in low-risk and obese populations. It is true that we acknowledge some limitations, but this finding also translates into postnatal metabolic benefits.
We have further clarified this issue in the Discussion section, by commenting on the significance of our findings, and potential explanations for them.
It will be very useful to compare the conclusions of the retrospective studies with those of your prospective study in the discussion section.
We appreciate the reviewer’s suggestion. WE have added some references on retrospective studies in our Discussion-Conclusions and commented them in the setting of our study (line 318):
- Absalom G, Zinga J, Margerison C, van der Pligt P. Associations of dietetic management with maternal and neonatal health outcomes in women diagnosed with gestational diabetes: a retrospective cohort study. J Hum Nutr Diet. 2019 Dec;32(6):728-736. doi: 10.1111/jhn.12682. Epub 2019 Jul 19. PMID: 31322776.
- Zhou S, Wang L, Chen J, Liu L, Wu X. Effects of individual dietary intervention on blood glucose level and pregnancy outcomes in patients with gestational diabetes mellitus: a retrospective cohort study. Ann Palliat Med 2021;10(9):9692-9701. doi: 10.21037/apm-21-2115
- Machado C, Monteiro S, Oliveira MJ; Grupo de Estudo de Diabetes e Gravidez da Sociedade Portuguesa de Diabetologia. Impact of overweight and obesity on pregnancy outcomes in women with gestational diabetes - results from a retrospective multicenter study. Arch Endocrinol Metab. 2020 Feb;64(1):45-51. doi: 10.20945/2359-3997000000178. Epub 2019 Sep 30. PMID: 31576966; PMCID: PMC10522280.
The results of a post hoc analysis should be viewed with considerable scepticism, and in my opinion, the degree of novelty regarding the benefits of a Mediterranean diet are well known. It would have been more useful to measure the CHO quantity per day in each patient.
We appreciate the reviewer’s comment. Even though our study entails the limitation of not specifically reporting CHO quantity per day in each patient, we believe that the uniformity of recommendations in the intervention group, and the questionnaires performed to evaluate compliance may serve as a surrogate marker.
However, we have acknowledged this limitation in the corresponding section of our final manuscript.

Reviewer 2 Report
Comments and Suggestions for Authors
in this manuscript authors investigated if Mediterranean diet (MedDiet) before 12th gestational week (GW) in women at high risk due to a BMI≥25kg/m2, could reduce the rate of GDM and metabolic syndrome (MetS) at 3 years postpartum. Authors found that the rate of GDM was significantly lower in MedDiet. Moreover, at 3 years postpartum, there was a reduction in the rates of impaired fasting glucose, obesity, waist circumference and metabolic syndrome.
the manuscript is interesting and generally well written. The topic is new and fit the aim of the journal. However, some points deserve to be improved.
Lines 41-46: It deserves to be pointed out that a high BMI before pregnancy is also associated with other pregnancy complications such as preterm birth (see PMID: 32102578 )
2.1. Study design: It should be added a table summarizing the diets composition investigated in this study
Tables: Statistically significant differences should be written in bold
4. Discussion: Authors should explain the results obtained more deeply. In fact, the beneficial effects of MedDiet on GDM and MerS occurrence could be due to anti-inflammatory and antioxidant effects of the diet itself. It deserves to be pointed out that GDM is charactherized by a chronic low grade inflammation (see PMID: 36359548) and inflammation plays a key role in insulin resistance and MerS occurrence (see PMID: 31781039). This is an interesting point to add since it can further highlight the important results obtained by the authors.
A graphical abstract would be useful
Abbreviations must be written in full length the first time that they are mentioned
Manuscript must be formatted according to the journal style (see the template)
Author Response
In this manuscript authors investigated if Mediterranean diet (MedDiet) before 12th gestational week (GW) in women at high risk due to a BMI≥25kg/m2, could reduce the rate of GDM and metabolic syndrome (MetS) at 3 years postpartum. Authors found that the rate of GDM was significantly lower in MedDiet. Moreover, at 3 years postpartum, there was a reduction in the rates of impaired fasting glucose, obesity, waist circumference and metabolic syndrome. The manuscript is interesting and generally well written. The topic is new and fit the aim of the journal. However, some points deserve to be improved.
Thank you very much for your constructive comments with which we agree. We have to make the following considerations
Lines 41-46: It deserves to be pointed out that a high BMI before pregnancy is also associated with other pregnancy complications such as preterm birth (see PMID: 32102578 )
We thank the reviewer for his/her valuable contribution. We have included this study in our reference list and commented it in the setting of our study (line 43).
2.1. Study design: It should be added a table summarizing the diets composition investigated in this study
We thank the reviewer for this suggestion. We have introduced in the Study Design section a paragraph (line 112) detailing the nutritional recommendations given to patients, since it is not a strict diet with a specific composition, but with specific recommendations:
The final text reads as follows:
“This was because they were all recommended to adhere to the principles of the MedDiet based on the use of EVOO as the only fat, in an amount greater than 4 tablespoons per day, and the consumption of a handful of nuts per day, in particular pistachios, uniformly in each study. In study 1, EVOO and pistachios were provided free of charge (10 L EVOO and 2 kg of roasted pistachios at 12-14 and 24-28 GW); whereas in study 3 only 2 kg of pistachios were provided free of charge at 12-14 and 24-28 GW. In study 2, only their consumption was recommended, but no food was provided free of charge. In contrast, women in the CG, both in study 1 and study 3, received identical recommendations to reduce oil consumption to less than 4 tablespoons per day, without the need of EVOO exclusively, and to avoid the consumption of nuts. The rest of recommendations were similar in both groups, such as: consuming 5 servings per day of fruit and vegetables, 2 or 3 servings of dairy products and prioritizing the consumption of whole grains, legumes, fish and lean meats. In addition, reducing the consumption of ultra-processed snacks, processed meats, commercial sweets and soft drinks was emphasized. It was also recommended to maintain an active lifestyle and use water as the main beverage.
IGs were compared with the two CGs in the RCTs. Women diagnosed with GDM were closely monitored by the Department of Endocrinology and received a consistent protocolized treatment, regardless of whether they were assigned to the control or intervention group.
During the study, pregnant women were followed-up uniformly to reinforce the nutritional intervention. Three visits were made during pregnancy, coinciding with the third, fifth and ninth month of gestation. At 3 months postpartum, another motivational interview was conducted to encourage all patients, regardless of the group to which they belonged, to follow the nutritional recommendations to liberalize the consumption of oil, preferably EVOO, and nuts, preferably pistachio, on a daily basis. After 3 years of the study, the patients were invited again for a voluntary follow-up visit to evaluate the results. However, several women refused to participate for different reasons, such as being pregnant again or health problems, among others.
Tables: Statistically significant differences should be written in bold
We have amended the text in the tables, as suggested.
- Discussion: Authors should explain the results obtained more deeply. In fact, the beneficial effects of MedDiet on GDM and MetS occurrence could be due to anti-inflammatory and antioxidant effects of the diet itself. It deserves to be pointed out that GDM is charactherized by a chronic low grade inflammation (see PMID: 36359548) and inflammation plays a key role in insulin resistance and MerS occurrence (see PMID: 31781039). This is an interesting point to add since it can further highlight the important results obtained by the authors.
WE appreciate the reviewer’s suggestions. We have further added this references and commented them in our final manuscript (line 368). The final text reads as follows:
Benefits of Mediterranean diet supplemented with EVOO and pistachios may be due to several mechanisms. For instance, it has been observed that EVOO reduces postprandial glucose levels and improves the inflammatory profile, and its use facilitates a greater intake of vegetables, traditionally consumed with EVOO in Spanish cuisine. It seems possible that it increases satiety due to the fact that fats delay gastric emptying and also thermogenesis. On the other hand, nuts, and in particular pistachios, are rich in unsaturated fatty acids, fiber, magnesium and other phytochemical components with possible beneficial effects on insulin sensitivity, fasting glucose levels and inflammation. Their antioxidant capacity is superior to other nuts, given their high levels of lutein, β-carotene and γ-tocopherol. Pistachio consumption improves inflammatory cytokine profiles related to the development of GDM
Imamura F, Micha R, Wu J, de Oliveira Otto MC, Otite FO, Abioye AI, et al. Effects of Saturated Fat, Polyunsaturated Fat, Monounsaturated Fat, and Carbohydrate on Glucose-Insulin Homeostasis: A Systematic Review and Meta-analysis of Randomised Controlled Feeding Trials. PLOS Med 2016; 13: e1002087. https://doi.org/10.1371/journal.pmed.1002087 PMID: 27434027
Schwingshackl L, Christoph M, Hoffmann G. Effects of Olive Oil on Markers of Inflammation and Endothelial Function-A Systematic Review and Meta-Analysis. Nutrients. 2015; 7:7651±75. https://doi.org/ 10.3390/nu7095356 PMID: 26378571
HernaÂndez-Alonso P, Salas-Salvado J, Baldrich-Mora M, Juanola-Falgarona M, Bullo M. Beneficial Effect of Pistachio Consumption on Glucose Metabolism, Insulin Resistance, Inflammation, and Related Metabolic Risk Markers: A Randomized Clinical Trial. Diabetes Care 2014; 37:3098±3105.https://doi.org/10.2337/dc14-1431 PMID: 25125505
A graphical abstract would be useful
We appreciate this suggestion. We have elaborated a graphical abstract which we believe improves the understanding of our study.
Abbreviations must be written in full length the first time that they are mentioned
We have amended the text according to the suggestion.
Manuscript must be formatted according to the journal style (see the template)
WE appreciate the suggestion. The final manuscript has been thoroughly revised and adapted to the Journal’s format.

Reviewer 3 Report
Comments and Suggestions for Authors
I have great pleasure in reviewing the article entitled Early Mediterranean-based nutritional intervention reduces the rate of Gestational Diabetes Mellitus in overweight and obese pregnant women: a post-hoc analysis of the San Carlos Gestational Prevention study.
Optimal type of nutritional intervention to prevent GDM in high-risk women is extreamly important, that's why this study is very interesting and could answer for clinical questions.
The given article is a compilation of three studies conducted on a similar group of female patients. In subsequent studies, there were different models for allocating patients to the study group and the control group, which significantly complicates the comparison of groups.
There are no specyfic infromations regarding allocation to IG and CG.
The article did not include information about the diet followed by the patients for 3 years after delivery
Moreover, there is no information regarding follow-up in there study groups.
The introduction to the article is very short.
Numerous tables are presented with a lot of data that is not used in the text of the article.
The literature is very limited.
Author Response
I have great pleasure in reviewing the article entitled Early Mediterranean-based nutritional intervention reduces the rate of Gestational Diabetes Mellitus in overweight and obese pregnant women: a post-hoc analysis of the San Carlos Gestational Prevention study.
Optimal type of nutritional intervention to prevent GDM in high-risk women is extreamly important, that's why this study is very interesting and could answer for clinical questions.
The given article is a compilation of three studies conducted on a similar group of female patients. In subsequent studies, there were different models for allocating patients to the study group and the control group, which significantly complicates the comparison of groups.
There are no specyfic infromations regarding allocation to IG and CG.
We thank the reviewer for his/her positive consideration on our study.
WE have further clarified the information on allocation to IG and CG (line 94 and 107). In this regard, participants were randomized in the 8th-10th GW by age, parity and BMI by an external investigator
The article did not include information about the diet followed by the patients for 3 years after delivery. Moreover, there is no information regarding follow-up in there study groups.
WE appreciate this suggestion.
In order to provide further details on these issues, we have performed the following changes:
In table 3, we provide data on adherence to nutritional follow-up, based on semiquantitative questionnaires.
All women were re-evaluated at 3 months postpartum, as usually recommended, and invited to participate in postnatal follow-up. All women received the recommendation to liberalize the consumption of oil, preferably EVOO and daily nuts, preferably pistachio in a motivational interview. Women were free to contact the research team and at least one annual visit with their family doctor for analytical-metabolic evaluation was recommended. A re-evaluation was recommended in our unit at 3 years postpartum.
The introduction to the article is very short.
We appreciate this suggestion. We have further developed the introduction section to set our study, and we have provided additional literature references.
Numerous tables are presented with a lot of data that is not used in the text of the article.
We appreciate this comment. We have provided full data in our tables to report the whole set of findings, but we have only commented in the text what we considered the most relevant findings, due to limitations in the extent of the manuscript.
We have briefly referred to some of the rest of data shown in tables, as suggested. Final text reads (lines 233): "However, there were no differences in other anthropometric variables, such as weight and length, nor in large-for-gestational-age (LGA) infants. There were also no differences in rates of prematurity or dystocia, umbilical cord pH, APGAR, or other adverse effects in the newborn…"
The literature is very limited.
We appreciate this comment. We have further included additional literature references, as suggested and others:
Absalom G, Zinga J, Margerison C, van der Pligt P. Associations of dietetic management with maternal and neonatal health outcomes in women diagnosed with gestational diabetes: a retrospective cohort study. J Hum Nutr Diet. 2019 Dec;32(6):728-736. doi: 10.1111/jhn.12682. Epub 2019 Jul 19. PMID: 31322776.
Zhou S, Wang L, Chen J, Liu L, Wu X. Effects of individual dietary intervention on blood glucose level and pregnancy outcomes in patients with gestational diabetes mellitus: a retrospective cohort study. Ann Palliat Med 2021;10(9):9692-9701. doi: 10.21037/apm-21-2115
Machado C, Monteiro S, Oliveira MJ; Grupo de Estudo de Diabetes e Gravidez da Sociedade Portuguesa de Diabetologia. Impact of overweight and obesity on pregnancy outcomes in women with gestational diabetes - results from a retrospective multicenter study. Arch Endocrinol Metab. 2020 Feb;64(1):45-51. doi: 10.20945/2359-3997000000178. Epub 2019 Sep 30. PMID: 31576966; PMCID: PMC10522280.
Zehravi M, Maqbool M, Ara I. Correlation between obesity, gestational diabetes mellitus, and pregnancy outcomes: an overview. Int J Adolesc Med Health. 18 de junio de 2021;33(6):339-45.
Paredes C, Hsu RC, Tong A, Johnson JR. Obesity and Pregnancy. Neoreviews. febrero de 2021;22(2):e78-87.
Giannubilo SR, Licini C, Picchiassi E, Tarquini F, Coata G, Fantone S, et al. First trimester HtrA1 maternal plasma level and spontaneous preterm birth. J Matern Fetal Neonatal Med. febrero de 2022;35(4):780-4.
Lloyd M, Morton J, Teede H, Marquina C, Abushanab D, Magliano DJ, et al. Long-term cost-effectiveness of implementing a lifestyle intervention during pregnancy to reduce the incidence of gestational diabetes and type 2 diabetes. Diabetologia. julio de 2023;66(7):1223-34.
Diaz-Santana MV, O’Brien KM, Park YMM, Sandler DP, Weinberg CR. Persistence of Risk for Type 2 Diabetes After Gestational Diabetes Mellitus. Diabetes Care. 1 de abril de 2022;45(4):864-70.
Teede HJ, Bailey C, Moran LJ, Bahri Khomami M, Enticott J, Ranasinha S, et al. Association of Antenatal Diet and Physical Activity-Based Interventions With Gestational Weight Gain and Pregnancy Outcomes: A Systematic Review and Meta-analysis. JAMA Intern Med. 1 de febrero de 2022;182(2):106-14.
Imamura F, Micha R, Wu JHY, Otto MC de O, Otite FO, Abioye AI, et al. Effects of Saturated Fat, Polyunsaturated Fat, Monounsaturated Fat, and Carbohydrate on Glucose-Insulin Homeostasis: A Systematic Review and Meta-analysis of Randomised Controlled Feeding Trials. PLOS Medicine. 19 de julio de 2016;13(7):e1002087.
Schwingshackl L, Christoph M, Hoffmann G. Effects of Olive Oil on Markers of Inflammation and Endothelial Function-A Systematic Review and Meta-Analysis. Nutrients. 11 de septiembre de 2015;7(9):7651-75.
Hernández-Alonso P, Salas-Salvadó J, Baldrich-Mora M, Juanola-Falgarona M, Bulló M. Beneficial effect of pistachio consumption on glucose metabolism, insulin resistance, inflammation, and related metabolic risk markers: a randomized clinical trial. Diabetes Care. noviembre de 2014;37(11):3098-105.
We appreciate this suggestion. WE have further added these references and commented them in our final masnucript.

Round 2
Reviewer 1 Report
Comments and Suggestions for Authors
The manuscript was improved according to the suggestions made.
Reviewer 2 Report
Comments and Suggestions for Authors
the manuscript has been significantly improved after revision and can be accepted in the current form